# Descriptive Epidemiology of Melanoma Diagnosed between 2010 and 2014 in a Colombian Cancer Registry and a Call for Improving Available Data on Melanoma in Latin America

**DOI:** 10.3390/cancers15245848

**Published:** 2023-12-15

**Authors:** Esther de Vries, Claudia Uribe, Claudia Catalina Beltrán Rodríguez, Alfredo Caparros, Erika Meza, Fabian Gil

**Affiliations:** 1Department of Clinical Epidemiology and Biostatistics, Faculty of Medicina, Pontificia Universidad Javeriana, Ed. Hospital Universitario San Ignacio, Carrera 7 N° 40–62–piso 2, Bogotá 110231, Colombia; fgil@javeriana.edu.co; 2Population Based Cancer Registry of Metropolitan Area of Bucaramanga, Universidad Autónoma de Bucaramanga, Calle 157 #14 55, Floridablanca 68276, Colombia; curibep@unab.edu.co (C.U.); cancerbmanga@unab.edu.co (E.M.); 3MSD Colombia, Calle 125 a 53–45 piso 8, Complejo Empresarial Colpatria, Bogotá 111121, Colombia; 4MSD Medical Affairs Latin America, Cazadores de Coquimbo 2841/57, Munro, Vicente López, Buenos Aires B1605AZE, Argentina

**Keywords:** cutaneous melanoma, surveillance, survival, stage, cancer registry

## Abstract

**Simple Summary:**

Melanoma is among the most aggressive types of skin cancer. For Latin American populations, we know very little about its patterns of occurrence and probabilities of survival. In this study, using data of the Bucaramanga Metropolitan Area Cancer Registry, we describe that the incidence rate is about 2 per 100,000 person-years, and that 5 years after diagnosis, 71% did not die of their melanoma. We also described the details of the characteristics and subgroups of survival. These findings help understand this disease better in this population and identified the need for more detailed and timely data in the whole region of Latin America.

**Abstract:**

We aimed to improve the available information on morphology and stage for cutaneous melanoma in the population-based cancer registry of the Bucaramanga Metropolitan Area in Colombia. The incidence and survival rates and the distribution of melanoma patients by age, gender, anatomical subsite, and histological subtype were calculated. All 113 melanoma patients (median age 61) were followed up (median time 7.4 years). This exercise (filling in missing information in the registry by manual search of patient clinical record and other available information) yielded more identified invasive melanomas and cases with complete information on anatomical localization and stage. Age-standardized incidence and mortality rates were 1.86 and 1.08, being slightly higher for males. Most melanomas were localized on the lower limbs, followed by the trunk. For 35% of all melanomas, the morphological subtype remained unknown. Most of the remaining melanomas were nodular and acral lentiginous melanomas. Overall global and relative 5-year survival was 61.6% and 71.3%, respectively, with poorer survival for males than females. Melanomas on the head and neck and unspecified anatomical sites had the worst survival. Patients without stage information in their medical files had excellent survival, unlike patients for whom medical files were no longer available. This study shows the possibility of improving data availability and the importance of good quality population-based data.

## 1. Introduction

The most reliable data source for the incidence rates or absolute numbers of new cancer cases in a population and associated information, such as population-based cancer survival, are population-based cancer registries (PBCRs) [1]. PBCRs are designed to capture all cases in a defined population (most frequently a geographical area such as a state or metropolitan area), with an emphasis on the use of the data to measure the burden of disease in the population: cancer patterns of the population and differences between them within subgroups, monitoring cancer trends over time and as such helping to collect information on the need for preventative, diagnostic and treatment facilities [2]. For PBCRs in many low- and middle-income countries, case identification and follow-up are the main challenges, and constrained resources often imply a focus on case identification and abstracting the main topographic and main morphology subtype of the cancer as well as patient demographic data, leaving other important variables such as stage at diagnosis and detailed morphology information as optional variables for registration [3,4]. 

However, such information is precisely of very high value in using the PBCRs to go beyond general incidence, mortality and survival reports and grasp the potential for improving early diagnosis, and that way, survival, etc. Such information is highly relevant to understand the population dynamics of cutaneous melanoma, a cancer type that is not very common in Colombia. The scarce literature on the topic seems to indicate that the epidemiology of melanoma in Latin America differs from that of the high incidence countries in North America, Europe and Oceania, with different incidence levels, different distributions by anatomical localization and morphology, and survival seems to be suboptimal, but a lack of detailed (population-based) data hampers a better understanding of the problem [5,6,7,8,9,10]). Overall, the limited available data inform us that age-standardized incidence rates of cutaneous melanoma are estimated to be around 3 per 100,000 person-years in Colombia and age-standardized rates vary between 1 and 5 per 100,000 person-years in Central and South America [5,6,7,8,9,10]. The scarcity of information on cancer incidence in Latin America in general, representing only between 3–10% of the population in the region [2], is exacerbated by the lack of information on stage at diagnosis, a variable that is very important to take into consideration when analyzing survival, but also an indicator of the level of timely detection [2,11]. 

For population-based cancer registries in South America, capturing information on stage at diagnosis is particularly difficult: many physicians do not describe the stage as such in the medical files, making the interpretation of clinical information necessary to determine the stage [11]. Additionally, for many patients, there was not enough information in the medical records to abstract the stage, not even in our specific exercise. Additionally, over time, medical histories go missing, making it impossible to retrieve this information years later.

The main objective of this study was to improve the available information on each melanoma case regarding the available information on the morphology and stage at diagnosis for cutaneous melanoma in the PBCR of the Bucaramanga Metropolitan Area, a region in Colombia, South America, covering 1.1 million inhabitants. In a special exercise, we revisited melanoma cases to complete this information. In this manuscript, we report the process and results of this exercise, and report on the resulting incidence and survival rates in this population-based cancer registry.

## 2. Methods

### 2.1. Population

As of 2002, the PBCR of Bucaramanga and its metropolitan area (BMA) in Santander, Colombia, collects the information of all of the residents of the Bucaramanga Metropolitan Area (Bucaramanga, Floridablanca, Giron and Piedecuesta—total population ~1.1 million) with newly diagnosed invasive cancers through visits to all healthcare and diagnostic institutions in the region [12]. The International Agency for Research of Cancer (IARC) have classified it as being of good quality, and its data have been included since the X volume of Cancer Incidence in Five Continents [4,13]. For this project, we included all patients that were newly diagnosed with invasive cutaneous melanomas between 2010 and 2014 (*n* = 113), defined as melanomas with ICD-O-3 topography codes 44.0 to 44.9, excluding other topography codes. The available information for the clinical variables in the standard cancer registry set was incomplete in many cases; we completed this missing information as much as possible by manually going back to the medical files and pathology reports—this search and the activity of completing the records is referred to in this manuscript as an “exercise”. The stage was recorded according to TNM characteristics and staged according to the 8th edition of the AJCC staging system [14]. 

The incidence date was defined as the date of issuing the pathology report, when available, and otherwise based on the first date of clinical diagnosis. All patients were followed up for vital status by linking with official data for at least five years, and the date of death was established by the “Resolution of Death” emitted by the National Registry of Civil Status (Registraduría Nacional del Estado Civil) and death certificates through the Colombian Bureau of Statistics (Departamento Administrativo Nacional de Estadísticas [DANE]). The quality of vital statistics is reasonable, if only the date of death is needed [15]. The date of last contact for those who are not deceased was the date of consultation of the various databases in Colombia which contain information on use of the health system (RUAF, FOSYGA and the database of the “comprobador de derechos” [checking of rights] of the departmental health secretary), which were accessed using the personal identification number of the patients [16].

As the data are from a population-based cancer registry which complies with the Colombian and international regulations and recommendations, no formal ethics committee approval was needed. Details on registry processes are provided elsewhere [12]. 

### 2.2. Quality Indicators

Cancer registries must provide an objective indication of the data quality. We adopted the four dimensions (comparability, validity, timeliness, and completeness) described by IARC and updated in 2009 [17]. Additionally, three key statistics influence data accuracy: the proportion of cases with missing data, the percentage of cases with a morphologically verified diagnosis (MV%), and the percentage of cases relying solely on information from a death certificate (DCO%).

Out of all cutaneous melanomas registered in this period, 108 were microscopically verified (%MV: 95.6%), 4 had a clinical diagnosis (3.5%) and 1 was based only on the death certificate. There were no cases with missing age at diagnosis. The mortality to incidence ratio was 0.66 for males and 0.5 for females.

### 2.3. Statistical Analysis

The distribution of melanoma patients by age, gender, anatomical subsite and histological subtype was analyzed.

Age- and sex specific incidence and mortality rates were calculated and standardized for age (age-standardized rates ASR) using the direct method, using the SEGI world standard population, and expressed per 100,000 person-years. 

Overall survival was calculated as time between diagnosis and date of death, using standard Kaplan–Meier methods, with right-censoring at lost to follow-up or administrative closure of the database. Relative survival was calculated using the Pohar Perme estimator for all 113 invasive melanomas defining survival time as the time between the date of diagnosis and the date of death, date of last contact, or date of the closure of follow-up (31 December 2021). We used regional life tables for each corresponding year of diagnosis for the Department of Bucaramanga from DANE. We stratified the analyses by sex, detailed anatomical site, histological subtype, and stage at diagnosis [16,18]. 

When there was missing information for the dates (exact date of birth, incidence date and date of last contact), detailed dates were imputed following these steps: when only the day was missing, it was imputed as the 15th keeping the month and year available; when the month and day were missing, they were imputed as the 1st of July, keeping the year available; the year of diagnosis was available for all cases in the dataset.

As the data represent a census of all melanomas diagnosed among the residents of the Bucaramanga Metropolitan Area, not a sample of the population, no confidence intervals around numbers and rates were calculated. 

Data management and data analysis were made using R software (version 4.2.1), relative survival analysis was made using the relsurv R package [19].

## 3. Results

### 3.1. Results of the Completion Exercise 

The details of the available information prior to and after the exercise are detailed in Table 1. Prior to the completion exercise, only 85 invasive cutaneous melanomas diagnosed between 2010 and 2014 had been identified. After the exercise, more invasive cutaneous melanomas had been identified, and the proportion of melanoma cases without detailed information on anatomical localization reduced from 13% to 8%, information on the complete 8th edition AJCC stage was completely absent prior to the exercise, and afterwards, was available for 35%, and T-stage was improved (0% information versus 67% with information). However, the proportion of cases with detailed information on morphology was reduced from 80% to 65%—indicating that the newly identified melanomas had less detailed information in their medical files. In 21 of the cases (19% of all melanomas), there was no information on stage available in the medical records, and in 52 cases (46%), the medical history was no longer available in the hospitals at the time of the data completion exercise (in 2021). This group of patients was treated as a specific category in the analyses. 

After completing the exercise, a total of 113 cutaneous melanomas were diagnosed in the Bucaramanga Metropolitan Area between 2010 and 2014, 55 in males and 58 in females. All 113 melanoma patients were followed up until the 31st of December 2021 through administrative cross-linking, 54 (48%) had died at the end of the follow-up, 59 (52%) were alive. The median age was 61 years (interquartile range 51 to 74). We did not observe differences by sex; therefore, the results are presented for both sexes combined.

All presented results regarding incidence, mortality and survival are based on the database after completing the exercise.

### 3.2. Population-Based Epidemiology of Cutaneous Melanoma in BMA, 2010–2014

#### 3.2.1. Incidence and Mortality

The crude incidence rates per 100,000 person-years of melanoma in the Bucaramanga Metropolitan Area in the studied period was 2.08: 2.12 for males and 2.04 for females; age-standardized incidence rates were 1.86 (2.0 for males and 1.67 for females). Crude and ASR melanoma mortality rates in Bucaramanga were 1.23 and 1.08, respectively (CR males 1.47, females 1.02; ASR males 1.41, females 0.83). 

Figure 1 provides a comparison of age-specific incidence and mortality rates in Bucaramanga, clearly showing an incidence appearing as of the mid-30 age-range with mortality closely following.

#### 3.2.2. Tumor Characteristics

Most melanomas were localized on the lower limbs (hips, legs, and feet), followed by the trunk. However, this hides the sex-specific differences in anatomic site distribution, where almost 1/3 of melanomas occurring in males were on the trunk, followed by the lower limbs and upper limbs, and almost 2/3 of melanomas occurring in women were on the lower limbs, followed by the trunk and head and neck (Appendix A). For 35% of all melanomas, the morphological subtype remained as NOS* after the completion exercise. Of the melanomas with morphological subtype information, most were nodular melanomas (NM), followed by acral lentiginous melanomas (ALM). When inspecting the frequencies of the morphological subtypes within anatomical localizations, NM were the most frequent subtype in the head and neck region and the trunk, whereas on the upper limbs, superficial spreading melanomas (SSM) and ALMs were the most frequent types and on the lower limbs, the ALM subtype prevailed. Lentigo maligna melanoma (LMM) was not mostly seen in the head and neck but rather on the trunk and limbs, where 80% was located (Table 2). 

Although the AJCC stage was missing for the majority of patients, some information could be recovered for tumor size, nodular involvement and the presence of distant metastasis in the TNM categories. For two-thirds of all melanomas, tumor size information (T) was available, with most melanomas being T1 (28%), but an important proportion was T3 or T4 (N = 20, 18%). However, for more than 2/3 of melanomas, no N or M information could be found in the medical records. Stage distribution was very similar between the sexes (results not shown).

While for some of the “unknown AJCC” stage, the tumor size (T) was known (for 32% (23/73) of the unknown AJCC stage was T unknown), for most the N stage was unknown (only 11 of 73 unknown AJCC cases had a known N-stage) and the M stage was completely absent (Table 1)

#### 3.2.3. Melanoma Survival 

Median follow-up time was 7.38 years (IQR 1.9–9.2 years), minimum 0.01, maximum 12 years. One case had zero survival time and was excluded from the analysis. 

Table 3 shows the 1 and 5-year overall and relative survival estimates. The overall (global) 5-year survival of invasive cutaneous melanoma in the Bucaramanga Metropolitan Area was 61.6%, with a 4.5 percent-point poorer survival for males compared with females. Relative 5-year survival was slightly higher with 71.3% and with the same higher survival pattern for women. Cutaneous melanomas on the head and neck and unspecified anatomical sites had the worst survival. Epitheloid cell melanomas had the lowest survival, followed by nodular melanomas and melanomas of an unspecified morphological subtype. 

Higher stage at diagnosis implied poorer survival. Patients without data on stage in their medical files had a high survival, unlike those patients for whom the medical file was no longer available. 

## 4. Discussion

Melanoma incidence in Bucaramanga is relatively low on a global scale, with age-standardized rates of around 2 per 100,000 person-years, like reports from other cancer registries in the region [5,8]. Incidence is low compared to high-incidence countries, slightly lower than the estimates for Colombia as a whole, but more or less average for South America (10). The mortality incidence ratio of 0.66 for males and 0.5 for females is high and indicative of the relatively poor survival of the melanoma patients (11).

Mortality rates in Bucaramanga are almost double those of the national data (Colombian national ASR for melanoma is 0.66: 0.71 for males and 0.61 for females) and reports from other regions [5], probably reflecting the average lighter skinned population in Bucaramanga despite the availability of good medical services and therefore diagnostic capacity. The completion exercise resulted in 28 cases having information on tumor size, morphology and detailed anatomical site; although information on complete stage remained limited, probably partially due to the unavailability of medical records upon revisiting the hospitals and private consultations. It seems quite consistent throughout the region to have, even after an intensified search for these data, an important proportion of cases with unspecified anatomical localizations (7–8% our data) and morphological subtypes (35% in our data). However, comparisons with previously published data of unspecified anatomical localizations (8.4–58.4% NOS) and morphological subtypes (48–99%) in Latin America show that this completion exercise and a focus on these variables results in substantial improvements [8]. 

Recently published data from Manizales, a city high in the Andean mountains with high smoking prevalence, showed that the 5-year survival rate for female melanoma patients was like for those in Bucaramanga, whereas male survival was much better in Bucaramanga. Comparisons between these studies should be made with caution, as the Manizales data also contained non-cutaneous melanomas, whereas we included only melanomas occurring on the skin. However, survival in both regions was poor compared to reports from high-incidence countries [5,20]

The most common body part for primary melanomas to occur was the lower limbs (for the total population and in women); however, for men it was on the trunk. This distribution is like what is observed in high incidence countries yet with a much more pronounced predominance of the lower limbs in the Bucaramanga population, probably because of the high proportion of acral lentiginous melanomas (ALMs) in South American populations, which occur mostly on the feet [5,6,8,9].

Unfortunately, despite the efforts to improve the available information, in 35% of the melanomas, the histological subtype was not known. Such large proportions of missing information make it difficult to really interpret these distributions. However, of the cases with this information, most were nodular melanomas (NMs) followed by acral lentiginous melanomas (ALMs)—which is unlike the distribution known in high-incidence, predominantly white populations, where the majority is superficial spreading and lentigo maligna melanomas [21,22]. 

The distribution by histological subtype was very similar to that of Manizales, except for superficial spreading melanomas (SSMs), which seemed relatively less common in our data (Bucaramanga 9.7% versus 17.4% in Manizales [5]). The greatest differences in histology compared to high-incidence countries are in the smaller proportions of SSMs or NMs and higher proportions of ALMs. In our data, some rare types of melanomas, such as amelanotic melanomas and epithelioid cell melanomas, were observed, which were not reported in other reports from South America [5,6,7,8,9,10]. 

The anatomical localizations of NMs, ALMs and Lentigo Maligna Melanomas (LMMs) are in some cases contradictory to descriptions in the literature: most NMs occurred on the head and neck and trunk, which is a distinct feature compared to higher incidence countries, where NMs are mostly located on the trunk. In our data, one ALM case was found in the head and neck region. This latter case was described as ALM by the pathologist but most likely represents a misclassified LMM in the head and neck region as acral lesions cannot occur on non-acral skin. Whereas LMM is in the literature described as almost uniquely occurring in the face [21], our data showed the majority of LMMs (90%) occurring on other body sites. These observations of unlikely localization/morphology combinations probably reflect difficulties in the correct application of histopathology criteria to reach a correct classification of the melanoma subtype by the pathologists. This, in addition to a frequent absence of clinical data, is a difficulty that pathologists in low-incidence regions face when evaluating pigmented lesions. 

The stage at diagnosis is an especially important variable, important for determining patient treatment and survival. Unfortunately, the stage is a notoriously difficult variable for cancer registries in Latin America to capture due to the frequent absence of such information in medical files [11]. In addition, the culture of recoding the stage is relatively new in the population-based cancer registries of Latin American cancer registries: many do not routinely collect stage at information—those that do seem to struggle with the same problems [5,11]. The specific search for stage information resulted in a substantial improvement in the available data for T-stage (41.4% in period 2000–2009 [9] versus 67% in 2010–2014). T information/Breslow thickness is generally available in the medical history/pathology report.

Identifying the information for the N and M stage, however, was notoriously more difficult, the N stage was only available for 27% of cases and the M stage was only available for 35% of cases. Crossing patients with unavailable AJCC with individual T, N and M information clearly illustrates that information on nodal involvement and the metastatic situation (N and M stage) was hard to find in the available medical records—indicative of the poor registration of this information in the medical records. The effect of the relative absence of the N and M stage is reflected in the very low proportion of patients with full AJCC staging, for which all components are necessary. The excellent survival of the patients without stage information in their medical files leads us to think that clinicians may be more inclined to write down stage information for patients with nodal involvement or metastasis—implying that the absence of N or M information likely implies N0 M0. Similar observations were previously made in high-incidence countries such as the Netherlands. Judging from the poor survival of patients without stage information because their medical files were no longer available, it is likely that many of them had advanced stages and that their files were no longer available because they had passed away [22]. 

The observation of a certain group of patients having a relative survival of more than 100% can be explained by the characteristics of relative survival, which is calculated by comparing the observed all-cause survival of the patient group to the expected survival of the population of origin in the same sex and age categories: if the particular patient group survives longer than the population of origin, the relative survival is more than 100%. Such excellent survival can be seen in groups that have cancer that can be treated easily with good a prognosis, and among a patient group which has a healthy lifestyle and good access to clinical care—characteristics which are likely to have occurred in these patient groups [23]. 

The advantage of the use of population-based data is that they, provided the registry has good coverage and data validity, provide unbiased information on cancer incidence without being limited to certain levels of socioeconomic status, ethnicity, etc. Unfortunately, the available data had no information on socioeconomic indicators nor phototype. The importance of population-based versus institution-based data, which more likely have more complete tumor-specific and clinical data, is clearly illustrated by two small South American data series on the melanomas of patients that attended private versus state clinics, with distinctly different distributions by anatomical site of occurrence and morphological subtype [6,7].

Survival of melanoma is, as in other parts of the world, highly determined by stage at diagnosis: whereas patients diagnosed in Stage I all survived the first five years after diagnoses, the 5-year survival for stage IV patients was only around 15%. As expected, males had worse survival than females [24,25,26]. 

### 4.1. Implications Regarding Data Collection by Cancer Registries

It is clear that the previously completely absent information on the AJCC, N and M stages and partially available on the T stage has been supplemented. The stage at diagnosis is a particularly important variable when analyzing prognosis and determines treatment options for the patient. However, it is a very difficult variable for population-based cancer registries to obtain in Colombia: the registry director informed (CU) “*we revised medical histories of the dermatologists, and they had no information on N and M, only on the tumor size; we also checked the pathology reports and some of them only contained the surgery reports without further staging data. Moreover, some oncological centers in the region have closed after 2014 and when we searched for the data in 2021–2022 it was not possible to obtain information anymore*”. For these reasons, stage information remained incomplete for most patients (65%). These results on the stage highlight the importance of registering the stage at the moment of the registration of the cases, but also the difficulties in obtaining this information and the importance of training physicians and adapting electronic medical records to facilitate T, N and M stage information in a standardized manner, considering these are no longer an option, but part of the “core variables” to collect. Having information for most patients in the cancer registries with this variable will enhance the usefulness of the cancer registries for clinical, public health and policy making questions to an important degree. 

As mentioned above, it is important to facilitate physicians in noting the nodal and metastatic information in the medical records to increase the proportion of melanoma patients with available information—facilitating dedicated spaces in the medical history for these variables could be of help. Similarly, the reporting of the morphological type should be improved. Therefore, this is a call for clinicians to help facilitate the collection of richer population-based cancer registry data by improving the availability of clear stage information in the clinical records. Additionally, working on simplified staging methods, such as the essential TNM, for cancer registry staff for all cancer types could help improve the availability of information in the cancer registry records [11].

The results also show the importance of timely data collection, to avoid difficulties in accessing medical records of deceased patients. At the same time, these data show the value of having population-based information on cancer incidence for melanoma and other neoplasms. These necessarily must come from population-based cancer registries, as other, more passive or administrative-based methods results in the over or underreporting of cancer cases. It would be desirable to increase the amount of population-based cancer registries in Latin America, now representing 3–10% of the entire population, to covering a proportion of about 20% of each country’s population, in strategic areas, to be able to monitor cancer incidence and other important indicators in the populations. National coverage is only feasible in smaller countries and data have shown that the extra investment in population-based cancer registries is very limited once about 20% of the population is covered [27]. 

### 4.2. Biases or Limitations

The advantage of population-based data is the absence of selection bias. However, the availability of detailed information on tumor characteristics is a problem. The lack of detailed stage information and on morphological subtype is unfortunate and limits the interpretation of, in particular, survival analyses for subgroups. However, the results suggest that when cancer registries decide to routinely collect this information at the moment of initial data abstraction, the amount of missing information can be reduced. 

Survival estimates heavily depend on the quality of the follow-up: missing deceased patients in follow-ups and censoring them artificially improves survival [16]. As in our study, follow-ups were performed by crosslinking with administrative data, for patients deceased in Colombia, the data can be assumed to be complete. However, patients who passed away in other parts of the world may have been missed. 

Although the Bucaramanga Metropolitan Area Cancer Registry has been evaluated as having decent quality indicators in terms of completeness and validity, it is always possible that some melanoma patients in the population were not registered: either because they were never diagnosed or because the cancer registry somehow missed this diagnosis. However, we believe the potential for this bias is limited. 

## 5. Recommendations and Conclusions 

The incidence of melanoma in the Metropolitan Area of Bucaramanga (Colombia) is relatively low, but mortality is high and survival poor by international standards. Adequate quality of information is essential to advance in research processes. In addition, it is necessary to emphasize the importance of the proper completion of clinical records, timely data collection, and the correct classification of pigmented lesions by the health care providers involved in this diagnosis. Our results show that late case identification or data completion may result in missing the stage, histology and anatomical localization information of patients whose medical records may no longer be available in institutions. The data clearly show that melanoma, although less common, does occur in Colombia and the population should be informed of the importance of protection from ultraviolet radiation through clothing, seeking shade and appropriate sunscreen use [28]. Physicians should be on the alert of melanomas, in an attempt to identify them in early, treatable stages. 

## Figures and Tables

**Figure 1 cancers-15-05848-f001:**
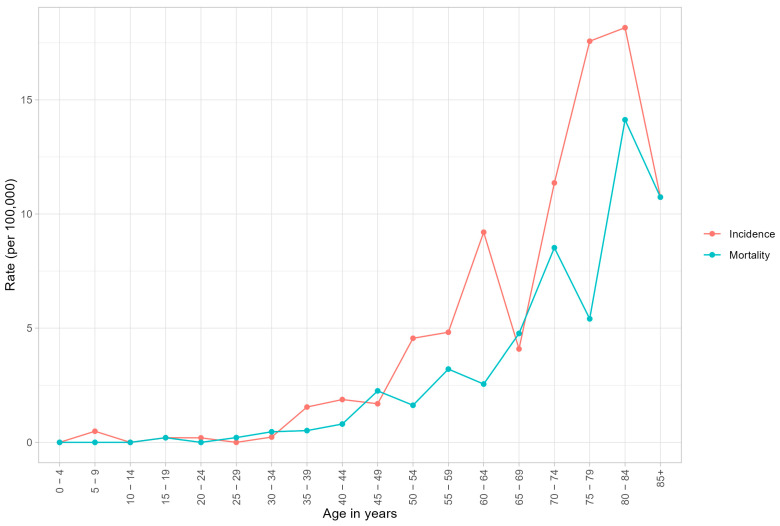
Age−specific incidence and mortality rates per 100,000 person-years in BMA: 2010−2014.

**Table 1 cancers-15-05848-t001:** Summary of the characteristics of the residents of the Bucaramanga Metropolitan Area diagnosed with a melanoma between 2010–2014.

Characteristic	Prior to Completion Exercise N (% of Total of 85) *	Final Sample (Exercise Completed) N (% of Total of 113)
**Anatomical subsite**		
Skin of head and neck	10 (12%)	17 (15%)
Lip	-	1
Eyelid	-	1
External ear	-	4
Other parts face	-	4
Scalp and neck	-	7
Trunk	23 (27%)	24 (21%)
Upper limb including shoulder	10 (12%	15 (13%)
Lower limb including hip	30 (35%)	48 (43%)
Skin not otherwise specified	12 (14%)	9 (8%)
**Morphological subtype**		
Amelanotic melanoma	0	1 (0.9%)
Superficial spreading melanoma	11 (13%)	11 (9.7%)
Melanoma of epitheliode cells	0	4 (3.5%)
Acral lentiginous melanoma	12 (14%)	21 (18.6%)
Lentigo maligna melanoma	18 (21%)	10 (8.9%)
Nodular melanoma	27 (32%)	26 (23%)
Not otherwise specified	17 (20%)	40 (35.4%)
**AJCC Stage**		
I	NA	3 (2.6%)
II	NA	0 (0%)
III	NA	2 (1.8%)
IV	NA	35 (31.0%)
Unknown—no stage information	NA	21 (18.6%)
Unknown—Medical history missing	NA	52 (46.0%)
T-stage		
T1	9 (11%)	32 (28%)
T2	30 (35%)	24 (21%)
T3	10 (12%)	9 (8%)
T4	22 (26%)	11 (10%)
TX or missing T	14 (16%)	37 (33%)
N-stage		
N0	NA	11 (10%)
N1	NA	9 (8%)
N2	NA	5 (4%)
N3	NA	5 (4%)
NX or missing N	NA	83 (73.4%)
M-stage		
M0	NA	5 (4%)
M1	NA	35 (31%)
MX or missing M	NA	74 (64%)

*: As mentioned in the excel file provided by the registry in during the construction of the protocol. NA = Not available.

**Table 2 cancers-15-05848-t002:** Distribution of morphological subtype within the localization of cutaneous melanomas in the BMA, 2010–2014.

	Anatomical Localization				
Morphological Subtype	Head and Neck N (Row %)	Trunk N (Row %)	Upper Limbs N (Row %)	Lower Limbs N (Row %)	Unknown ** N (Row %)
Malignant melanoma, NOS * (except juvenile melanoma)	4 (10%)	9 (23%)	3 (8%)	16 (40%)	8 (20%)
Nodular melanoma	9 (35%)	9 (35%)	1 (4%)	7 (27%)	0
Amelanotic melanoma	0 (0%)	0 (0%)	0	1 (100%)	0
Lentigo maligna melanoma	1 (10%)	3 (30%)	2 (20%)	3 (30%)	1 (10%)
Superficial spreading melanoma	0 (0%)	3 (27%)	5 (45%)	3 (27%)	0
Acral lentiginous melanoma, malignant	1 (5%)	0	3 (14%)	17 (81%)	0
Epithelioid cell melanoma	2 (50%)	0	1 (25%)	1 (25%)	0
**Total**	**17**	**24**	**15**	**48**	**9**

* (Not Otherwise Specified), ** Unknown localization.

**Table 3 cancers-15-05848-t003:** Relative survival (using the Perme method) of melanoma patients diagnosed in the BMA cancer registry between 2010 and 2014.

	Number of Patients	Crude (Overall) Survival		Net (Relative) Survival	
		1 y	5 y	1 y	5 y
**All patients**	113	87.5%	61.6%	89.7%	71.3%
**Sex**					
Male	55	88.9%	59.3%	91.1%	69.3%
Female	58	86.2%	63.8%	88.4%	73.2%
**Topography**					
Head and neck	17	88.2%	47.1%	90.1%	55.7%
Trunk	24	87.5%	54.2%	89.1%	59.9%
Upper limbs	15	86.7%	86.7%	89.0%	103.0%
Lower limbs	48	89.6%	64.6%	92.4%	75.5%
Skin, NOS *	9	75.0%	50.0%	75.9%	54.9%
**Morphology**					
Malignant melanoma, NOS *	40	71.8%	53.9%	72.9%	60.1%
Nodular melanoma	26	96.2%	53.9%	98.8%	66.9%
Amelanotic melanoma	1	100.0%	-	101.8%	-
Lentigo maligna melanoma	10	100.0%	70.0%	102.0%	75.2%
Superficial spreading melanoma	11	100.0%	100.0%	101.3%	108.3%
Acral lentiginous melanoma	21	95.2%	71.4%	99.5%	86.3%
Epithelioid cell melanoma	4	75.0%	25.0%	77.2%	38.1%
**Stage**					
I	3	100.0%	100.0%	101.5%	109.3%
III	2	100.0%	100.0%	102.2%	114.1%
IV	35	64.7%	14.7%	66.0%	17.1%
No data in clinical history	21	100.0%	100.0%	102.2%	115.4%
No clinical history	52	96.2%	73.1%	99.0%	85.1%
**by T**					
1	32	90.6%	71.9%	93.4%	84.0%
2	24	91.7%	75.0%	93.6%	83.7%
3	9	77.8%	44.4%	80.4%	60.1%
4	11	100.0%	45.5%	101.9%	52.0%
Unknown	37	80.6%	52.8%	82.4%	60.6%

* NOS = Not otherwise specified.

## Data Availability

The data presented in this study are kept under custody of the Bucaramanga Metropolitan Area Cancer Registry and are available on request from the director of the registry: cancerbmanga@unab.edu.co.

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
