# Peer review of "Descriptive Epidemiology of Melanoma Diagnosed between 2010 and 2014 in a Colombian Cancer Registry and a Call for Improving Available Data on Melanoma in Latin America"

_cancers, 2023, doi:10.3390/cancers15245848_

Round 1

Reviewer 1 Report

Comments and Suggestions for Authors

Can the authors please discuss why they did not look at pathology or radiology reports to get to staging of the tumors?

Since the authors state that the incidence of melanoma in Bucaramanga is low then what are some potential causes of higher mortality?

Reviewer 2 Report

Comments and Suggestions for Authors

The Authors performed a study about epidemiology of melanoma in Colombia. The article is of interest, however, some changes are needed:

- In the Introduction please add some information about the actual incidence of melanoma i Colombia and Latin America

- Please, always in the Introduction, stree about the pressing need to have a national registry and maybe a melanoma registry for Latin America

- I do not have observations about Methods

- In the results there are no information about Skin phototype, please add this information in the results

- No case of amelanotic/hypomelanotic melanoma, maybe is this due to the general phototype? Please specify

- Please, specify what do you mean with "epithelioid cell melanoma", since this term is not always confident with all the readers.

- In the Discussion please add some sentences about the important role of sun-protection and sunscreen SPF30-50 also in Skin phototypes usually present in Latin America

- In the Discussion, please speculate about the role of vitamin D in melanoma according to anatomic site and specifically shield sites melanoma or not. Accordingly read and report in references this article "Clinicopathological features, vitamin D serological levels and prognosis in cutaneous melanoma of shield-sites: an update. Med Oncol. 2015 Jan;32(1):451. "

- Please, stress in the Conclusions your findings.

Thank you.
